# Baron Eric Hermelin—Translation and the Merge of Traditions; Encoding and Reception of Persian Sufi Poetry in 20th Century Sweden

Rikard Friberg von Sydow 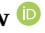

Historical and Contemporary Studies, Södertörn University, 141 89 Huddinge, Sweden; rikard.friberg.von.sydow@sh.se

**Abstract:** The mysticism of the 19th and 20th centuries has often been perceived as a reaction towards the new fast, dense, and modern world. But is it not so that it thrives on the same material foundations, globalism, networks, and mass production of text, that built our contemporary global information society? In this article, thoughts found in the writings of the Swedish mystic and translator Baron Eric Hermelin are analyzed. Hermelin was born into a Swedish noble aristocratic family in 1860. After traveling through the British Empire as a soldier in his youth, he returned to Sweden with books and knowledge. Unfortunately, he spent most of his remaining life incarcerated in the mental hospital St. Lars in the university town of Lund in the south of Sweden. But from the hospital, he released translations of Rumi, Khayyam, and other Persian mystics as well as reflections on Böhme and Swedenborg. The analysis will use Eric Hermelin's work and focus on the process of creating and delivering the texts of the Persian Sufis to a Swedish audience.

**Keywords:** mysticism; translation; Eric Hermelin; encoding

## 1. Introduction

The mysticism of the 19th and 20th centuries has often been perceived as a reaction towards the new fast, dense, and modern world (Borchert 1994, p. 306). But is it not so that it thrives on the same material foundations, globalism, networks, and mass production of text, that are actually an important part of this modern world? This serves as a starting point when analyzing processes and structures found in the published writings of the Swedish mystic and translator Baron Eric Hermelin. Hermelin lived from 1860 to 1944 and was born into a Swedish aristocratic family. After traveling through the British Empire as a soldier in his youth, he returned to Sweden with books and knowledge. Unfortunately, he spent most of his remaining life incarcerated in the mental hospital St. Lars in the university town of Lund in the south of Sweden. Even from the hospital, he was able to release translations of Rumi, Khayyam, and other Persian mystics as well as translations of Christian mystics such as Böhme and Swedenborg.

The main focus of this article will be on analyzing Eric Hermelin's work and exploring the process he used to create and deliver the texts of the Persian Sufis to a Swedish audience. What could the consequences be when such delivery of a phenomenon to a cultural context is performed by a single individual? What happens with a phenomenon when its reception in a new cultural context is deeply connected to this individual and the compilations of his works? This article provides an overview of Eric Hermelin's work, but it should be noted that there is much more to be explored. The analysis presented here only covers a small portion of Hermelin's extensive corpus and is primarily based on examples. The purpose of this overview is to lay the groundwork for further research in this area.

*Eric Hermelin's Life and Legacy*

Eric Hermelin was born in 1860 into a Swedish noble family. Following a brief stint as a student at Uppsala University, he was sent abroad by his family who was concerned that his unconventional behavior and penchant for heavy drinking would lead to scandal. (Lindahl 1982, p. 39). His travels took him to North America, where he found faith through preachers of born-again Christianity, to Great Britain where he eventually joined the Imperial army and was shipped to India, and finally to Australia, where he apparently behaved in such ways that he was sent home and put in a mental institution. There, at St. Lars Psychiatric Hospital in Lund, a university town in the south of Sweden, he used language skills he had acquired during his stay in India to translate a great amount of Persian medieval Sufi poetry into Swedish. Being stationed in Quetta in contemporary Pakistan, he had the possibility to pick up the language, and by using dictionaries, he could pick up parts of the language he did not have the time to learn earlier (Lindahl 1982).

Hermelin's work ethic, alone at the mental institution, was flawless. By 1935, he had translated 20 volumes of Persian Sufi poetry into Swedish (Ekerwald 2016, p. 9). His translations were also published. His first short pamphlet was published in 1913. In 1918, his first book was published by the major Swedish publisher P.A Norstedt after a recommendation by both the archbishop of the Swedish Church Nathan Söderblom and the famous Danish critic George Brandes (Peterson 1998, p. 42). The last published text arrived in 1943, one year before Hermelin's death. During this period of 30 years, he translated and published 33 volumes and seven smaller publications with the help of various Swedish publishers (Ekerwald 2016, p. 237). During the 1940s, he gained some fame outside more closed circles, and his translations were broadcast on Swedish National Radio (Falu Länstidning. 14 June 1940). His eightieth birthday was announced in the major Swedish newspaper *Svenska Dagbladet* in 1940. Hermelin is described as a prominent connoisseur of old Persian poetry with unique knowledge of language and culture gained through travels and through his aristocratic upbringing (Svenska Dagbladet. 20 June 1940). In the end, Hermelin is the main reason that no other language in the West has so much Persian Sufi poetry in translation as Swedish (Ekerwald 2016, p. 27).

Eric Hermelin had fierce opinions regarding the political events of his time, although they changed during his life. In the tradition of, among others, William Blake, he used his commented translations as a political and religious tool (Borsuk 2018, p. 121f). He started as a fierce conservative, supporting a known Swedish campaign to acquire a modern warship in 1912. During the First World War, he was supportive of Germany. However, his views evolved over time. He was very worried about the development in Germany during the 1930s, especially with the rise of antisemitism. All his books published after 1939 were sold for the benefit of Jewish people fleeing from the Third Reich (Ekerwald 2016, p. 15). This put him in a peculiar position as a Swedish conservative, a political sphere in which many agents leaned towards the Third Reich and antisemitism during this period of time. This is a position that might have been peculiar during the 1930s, but made him a person with decent opinions during a dark time, in the era after the Second World War (Aftonbladet. 3 February 1959). Taking into account Eric Hermelin's unique worldviews and opinions, as well as his prominent position as the primary translator of the Persian Sufi corpus in Sweden, he can be seen as a gatekeeper for the Swedish reception of Persian Sufism.

## 2. Information Flow and the Gatekeeper

To analyze Hermelin's role as a gatekeeper, two different theoretical foundations with related methods and techniques are used. A primary theoretical foundation can be called "information flow", and it is based on a blend of information flow analysis and process management commonly utilized in the field of archival science. The other theoretical foundation is rooted in reception theory, a theoretical orientation connected to a wide variety of disciplines from theology to media science and aesthetics. Information flow aids in answering questions regarding how information entities move from one location to another and what they pass on their way to or through a phenomenon. This flow

can be performed in various ways. In a society, such as a contemporary society, with a very dense layer of information and many nodes sending information back and forth, the description would be of a rhizomatic flow, a "patch" of information transfer involving several transferring nodes and a mutual exchange of information back and forth. Patches as a concept have been used to describe transfers in an ecological context, but also to describe the transfer of information in a human social context where a large number of agents are involved—such as the transfer of information regarding mushroom gathering (Lowenhaupt Tsing 2017, p. 62). It is possible to describe this rhizomatic flow as a more advanced version of a process—a process being a simple one-way directed flow.

In the case of Eric Hermelin, the translation of Persian Sufi poetry and the reception of these translations in a Swedish context, a theory based on a process of information is sufficient enough as an explanatory tool. There were very few persons with a knowledge of the Persian language, an interest in Sufi poetry, and enough time on their hands to perform similar translations. Nobody else translated Sufi poetry to Swedish during the period that Hermelin was active. Or if they did, their work was not published. Eric Hermelin had a monopoly on the market of translated Persian Sufi poetry in Sweden during his life, and also, today. There are a few other fragments of translations published during his time by other translators. Willy Kyrklund, a prominent Swedish writer with some knowledge of Persian, translated some verses of the *Mesnavi* in a text about his travels in Iran, but Hermelin is the only one who aimed towards translating a total corpus of texts (Kyrklund 1965, p. 82ff; Hållén 2013).

There are two tools that will be used in this analysis. The first is information flow, which can be visualized with the process—a concept often used in disciplines such as business administration and archival science. A process can be described as follows:

> *"A process is about managing entire chains of events, activities, and decisions that ultimately add value to the organization, and its customers. These chains of events, activities, and decisions are called processes."* (Dumas et al. 2017, p. 1)

In this case, we are observing a process that is a bit different from the above description. It is still a chain of events, activities, and decisions, but the parts concerning "value", "organization", and "customers" must be modified. The process being described here encompasses more than just a single organization and involves both smaller agents (individuals) and larger entities (publishers, magazines). We can discuss if it adds any value, not in the way that value is added in a commercial organization in any case. Regarding the customers, of course there will be a customer in the end. Books are published with the hope that somebody will buy them. The most important part here is thus a chain of events performed by different agents—a chain of events chosen here to be called a process. But we need to add two other concepts to obtain a better view: the environment and the gatekeeper.

The environment in which Eric Hermelin carried out his translations differed significantly from the context of the present day. It could be argued that these early threads of our current information society are present in Hermelin's work. The emergence of a form of globalism is evident, wherein an individual, by aligning themselves with the colonial powers as Hermelin did, was able to travel the world. This interconnected world also created the possibility of an interchange of information in the form of books, which made it possible for a translator, located in a mental hospital, to acquire printed texts of old writings in their original language. Hermelin's connection to the British Empire, and its colonial army, is interesting in regard to these environmental factors. The British Empire of the 19th and early 20th centuries has been named the first information society where the movement of information was as important as the movements of troops. The researcher Thomas Richards, in his book The Imperial Archive—Knowledge and the Fantasy of Empire, described how this need for collecting information infiltrated everything from military intelligence to culture and led to an accumulation of information that was greater than the possibility to process it (Richards 1993). When Eric Hermelin entered the service of this empire, he docked himself to an organization with a global reach that put him in

contact with contexts, languages, and information flow that he would not have had as big possibilities to come in contact with without him being recruited to the colonial army.

There are additional factors at play that position Eric Hermelin as a gatekeeper of Persian Sufi poetry within the Swedish market. One is that very few had the global experience that he had, in combination with an interest in mysticism, and all the time in the world. In a more deeply interconnected world, the one present today, there is a larger possibility that more than one person has these experiences. In modern times, the same devices utilized for daily tasks such as paying bills, watching television, and working can also provide access to texts from around the world. The possibility to be the only one with certain interests and connections is much smaller today. During Hermelin's time, this was much more likely. This characteristic is what positions him as a gatekeeper, and the following section will provide an explanation of what gatekeeping entails in this particular context.

A process consists of different agents performing acts that make the process go forward from beginning to end. A gatekeeper is an agent that performs a specific task in this process, a task needed to make the process go forward. It could be interpreted as a "keeper of a gate" with a negative connotation. But in this text, the connotation is supposed to be neutral. Eric Hermelin is a gatekeeper—but not through any monopoly—the only monopoly being his eccentricity and his, in Sweden, unique interest and ambition. The gatekeeper is created not through force but through obscurity—created by chance through a situation in which only one person has the knowledge and motivation—and in the end has his translations published. Based on the sources available during the research, it appears that Eric Hermelin was the sole translator and commentator of Persian Sufi texts, as no other major translator has been identified through searches of the National Library Catalog and contemporary news media.

Where does this process start? There are different possibilities of choice regarding a starting point—at least it should start before the gatekeeper enters the process. Regarding the end of the process, it is plausible to claim that it ends in the public's reception of the translated corpus. Reception theory fits well here, and it will be implemented in two ways: through analyzing reviews and articles written during the middle of the twentieth century and through analyzing the encoding in which Sufi poetry was served to the Swedish public.

Stuart Hall introduced the concept of encoding with his seminal article "Encoding and Decoding in the Television Discourse" (Hall 1973). According to Hall, encoding is what happens before content is released. It is the context and the ideology added to the content by the publisher—knowingly and unknowingly. There are several factors involved here: the limitations of the technical format used to distribute the content—in Hall's example, television, and in the presented case, printed matter; the structure of production—regarding Hermelin, his translation done in solitude in a mental hospital—but also the publishing process with everything that is included in such; last but not least, "the framework of knowledge" in which the ideological part is included—the creator of meaning. To create a transfer between the producer and the consumer, there needs to be some kind of symmetry between the encoding of the producer and the decoding of the consumer. Then the transfer creates a meaningful discourse (Hall 1973, p. 5ff). Hall's theory is connected to a setting in which a TV program is broadcast and consumed at the same time—and this during a time with fewer reruns and small possibilities to record a program for later reconsumption. Literature works in a different way. It has a greater possibility to ripen over time. But the concepts of encoding and decoding constructed by Hall are still plausible to use regarding printed text. Today, television has bridged this gap through the technical gains of the information society. In this article, encoding will be used to describe two processes. First and foremost is how Eric Hermelin creates a package out of Persian Sufi poetry that is customized for himself and for a Swedish context. Further on in this text, the reviews and repackaging of Hermelin's work are shown to follow a similar pattern.

There are of course other descriptions of how a cultural influence hits a cultural sphere. One such description is Simon Goldhill's book *Who Needs Greek?* which shows how the

rediscovery of Ancient Greece was handled by the West from the sixteenth to the twentieth century (Goldhill 2002). This is of course a much greater movement than what happened when Hermelin translated the Persian Sufi, but it follows the key claim in this text: that the persons involved in the rediscovery will reinterpret and situate the information which a cultural influence consists of into the context which they themselves are a part of.

The investigation will now follow in two parts. First, in Section 3, an analysis of two of Eric Hermelin's published translations, the *Shahnameh* and the *Mesnavi* is performed. The focus of this analysis is on annotation and references added by Hermelin to the original texts. This section ends with a passage in which the occasional appearance of Hermelin and his translations in the Swedish press during the time he was active as a translator and interpreter is described. Section 4 describes the legacy of Hermelin after his death, with a focus on the compilations of his work that were published from the 1970s onward.

### 3. Consequences of Eric Hermelin as a Gatekeeper

This investigation is not an evaluation of the quality of Hermelin's translations per se. According to both contemporary experts such as H.S. Nyberg, professor of Semitic languages at Uppsala University, and later experts such as Bo Utas, professor of Iranian languages with the same alma mater, Hermelin's translation held high quality. H.S. Nyberg used the translations for his students, and Bo Utas has claimed that they are almost too exact, thus losing some poetic qualities (Ekerwald 2016, pp. 27, 81; Utas 2011, p. 63). This might be the case because Hermelin was a keen user of dictionaries, being isolated from both a Persian and an academic setting, where a discussion regarding the use of certain words could be discussed. There are discussions in his references regarding what consequences different interpretations would result in, which shows an awareness of his role as a translator (Rumi 1934b, pp. 6, 20).

There are also some peculiarities in his use of the Swedish language that should be noted. Hermelin despised the Swedish spelling reform of 1906 and the subsequent translation of the Bible in 1917 (Hermelin 1999, p. 19) This makes his Swedish rather archaic, even for his time. He is also a keen user of hyphens, often splitting words that are commonly not divided in Swedish such as "Bröd-gifvaren"—bread giver—usually written "brödgivaren" and "Väg-visaren"—Guide—usually written "Vägvisaren" (Firdausi 1931, p. 5). If he discovers part of a text with an erotic undertone, he presents this part not in Swedish but in Latin, thus closing the door to these themes for what he considered the less intellectual (or less mature) reader (Hermelin 1999, p. 117). These are descriptions of encoding that are very close to his role as a translator, affecting the language through which the translation is presented. The language-inflicting encoding is quite easy to explain and appeals not to any advanced further analysis. The encoding which includes the addition of content (the translator notes to the text) and the compilation of Persian Sufi texts together with other texts of Western origin implies further analysis.

There are several examples of such encoding in Hermelin's translations. Here they are divided into two categories. One is called anachronism, in which references to contemporary (to Hermelin's time) events, political debates, and wars are categorized. These references are anachronistic because they mix together events and debates from Hermelin's contemporary time with medieval material created in another context. The other category is called "references to unrelated mystical writings" in which the connections (through, for example, notes) made to Bible verses, Swedenborg, Böhme, and so forth are categorized. These could also be described as anachronistic, but in relation to a specific theme, and thus are in need of a category of their own. Performing this analysis, the references in which Hermelin discusses his translation, his use of words and comparisons to other translations, are ignored. This analysis is performed by using two different texts translated by Hermelin—Firdausi's *Shah-Namah* (published in 1931) and Jalas al-Rumi's *Mesnavi* (published 1933–1934).

### 3.1. Firdausi's Shah-Namah

Hermelin's translation of Firdausi's *Shah-Namah* was published by Nordstedt in 1931 ([Firdausi 1931](#)). *Shah-Namah* is not a Sufi-related text, but rather an epic poem written by the poet Firdausi around 1000 AD. But it has been encoded by Hermelin as a mystic text and incorporated into his corpus. This is not done completely ad hoc by Hermelin. *Shah-Namah* (today usually transcribed as *Shahnameh*) is considered to be inspired by Sufi writings being a result of earlier heroic epos later influenced by Sufi poets ([Atooni 2022](#)). As an encoding to his translation, Hermelin adds notes and informative parentheses. Most of these are regarding biblical verses he claims are connected to or in some way related to the content. One example is as follows:

*Hvar och en, som slår Jernet (Jer. 23:29) mot (Hez. 36:26) en sten,*

In this case, it is a verse ("Everyone who hits the iron against a rock") from *Shah-Namah* ([Firdausi 1931](#), p. 21) connected to Jeremiah 23:29 ("'Is not my word like fire,' declares the Lord, 'and like a hammer that breaks a rock in pieces?'") and Ezekiel 36:26 ("I will give you a new heart and put a new spirit in you; I will remove from you your heart of stone and give you a heart of flesh."). This is one of many examples of the use of Bible verses—in some parts of the poems, these parentheses with Hermelin's comparisons of other texts are almost as common as the translated text itself. These references to Bible verses can be more or less obvious in their relevance to the reader. There might be a reason for these seemingly ad hoc additions to the translation, and they will be analyzed further later in this text.

In the part of *Shahnameh* describing the regime of King Hushang, the text offers one of these examples that could be described as less obvious to the reader. In the description of Hushang, there is a time during his life when he is practicing the profession of being a blacksmith. Here Hermelin makes a reference to the second chapter in the book of Zechariah, usually referred to as the story about "the man with the measuring stick" who measures the size of the future Jerusalem. To the reader, there are at least two interpretations of this reference. One is more practical—Hermelin shows us two examples of similar craft-related professions. The other interpretation is related to a theological explanation. The man with the measuring stick is doing something else than the merely practical; he is measuring a future heavenly (or God-given) kingdom. In this case, the reference, when it is encoded into the context of Hushang, would claim something else than just a king practicing a practical profession. Suddenly this blacksmithing is not at all practical crafting. It is claimed to be related to a transcendental sphere, being the creation of something else, something related to God—maybe a God-given task of creation. This is of course my interpretation; to the reader in general, Hermelin's reference—especially today, when the translator cannot explain his choices—will be a mystery for which it will be hard to find a more exact explanation.

There are 140 references to Bible verses from both the Old and the New Testament in Hermelin's 87 pages of translated text from the *Shahnameh*. Most of them follow a certain path that is easier for the reader to understand than the examples above. The most common are the description of Kings—which have a high occurrence in *Shahnameh*. Here the references follow the pattern that can be described in the following way: "The king X did Y" (*Shahnameh*)—"reference to a Bible verse where another king is described in a similar fashion or doing a similar activity". These references can be explained as a help to the reader (in a Christian context) to relate the text to other texts that he or she might be more familiar with. But it is also a method for the translator to encode the text into a certain setting in which these similarities have meaning. I will return to a discussion regarding this—the claim of common mystical roots—in the next part of the article regarding the translation of Rumi's *Mesnavi*.

There are other references made as well to a great variety of other texts other than the Old and the New Testament. These can be described as very varied—some being related to the contemporary setting that Hermelin worked in. Others could be described as striving to connect the *Shahnameh* to a Swedish setting. Examples of references that connect the translation to a Swedish setting are the Swedish medieval poem "Engelbrektsvisan" and

other epic Swedish poetry such as "Frithiof's saga", Esaias Tegner's modern retelling of the Icelandic fornaldrasaga bearing the same name. It seems like this is a way for Hermelin to present his translation to a Swedish reader, showing that the descriptions of kings and battles are similar to those that the reader might have read before. There seems to be no religious motivation to this—the examples are not connected to mysticism, more to Swedish epic writing with some slight similarities to the *Shahnameh*.

Other references are to the Swedish poet Gustaf Fröding and the British writer Robert L Stevenson and his famous novel *Dr Jekyll and Mr Hyde*. These are two references more connected to Eric Hermelin himself as a translator, or really, the person behind the translator. He had for a long period of time been very inspired by Stevenson and considered the story of the doctor with an alternative personality very close to himself. This might be one of the reasons why he chose London as a destination when he had to travel abroad—the novel takes place in London. When Hermelin enlisted into the British colonial army and had to construct a fake name for his service, he chose Edward, which also is the first name of Mr Hyde (Lindahl 1982, p. 56). Gustaf Fröding, the Swedish poet, was born the same year as Eric Hermelin, and they studied in Uppsala during the same period of time. Their life stories are to a certain degree similar; both suffered from alcoholism and mental illnesses and ended their lives in mental institutions. According to the biographer Per-Erik Lindahl, Hermelin found a soulmate in Fröding through his poetry, and he is, which I have discovered too, often used as a reference and as an example for comparison in Hermelin's translations (Lindahl 1982, p. 83).

There are also several references to Jakob Böhme and Emmanuel Swedenborg. The references to Böhme and Swedenborg are rather constant through Hermelin's translations, and I will return to discuss them in the analysis of the translation of Rumi's *Mesnavi* below. There are also references to the female British author Vernon Lee and her work on pacifism *Satan the Waster*. From Lee's book, Hermelin lifts a quotation about nations being fooled into wars and adds this to the part of *Shahnameh* in which the emperor Dhokak, who by the writer is portrayed as an evil war-monger in the epos, is described (Firdausi 1931, p. 51). *Satan the Waster* was published in 1920 and could have reached Hermelin while he was working on the translation of *Shahnameh*. Vernon Lee was, like Hermelin, a critic of modernism and modern life (Gagel 2019).

### 3.2. Jalal al-din Rumi's Mesnavi

Eric Hermelin's translation of Jalal al-din Rumi's *Mesnavi* was published in both a collected volume called *Mesnavi* (Rumi 1933) and in two excerpts called *Persiska låtar I–II* (*Persian Songs I–II*) (Rumi 1934a, 1934b). These excerpts are commented on differently than the full text and contain different notes and appendixes, so in this analysis, I will use all three texts, although they overlap. It seems, although it is very hard to prove, that Hermelin included the comments and references that the publisher did not want to include in the full text into the two smaller pamphlets that the *Persian Songs* consists of. This might have been the result of a negotiation between the translator and publishers. Eventually, this question could be answered by archival research, but that is not within the scope of this article.

In the version which includes the complete *Mesnavi*, the references to other texts are mostly done by stating the pages in texts in which Hermelin sees similar or parallel messages. In the two excerpts though, the references are long and detailed, sometimes including portions of text that are more developed than the actual translations. It is obvious that Hermelin is not only the translator of these publications; he is also the main interpreter. There are some small parts of theology happening in these texts too, and not just through the translator comparing and referencing different texts. Hermelin uses the references to define mysticism as a trinity of "reality", "supersensitivity", and "truth" (Rumi 1934b, p. 65). He makes this definition in reference to a short verse in which Rumi states that for the follower of mysticism, the death of the body is a gift. "Reality" is the only word that actually exists in the original text.

What Hermelin seems to touch here is part of a greater discussion of what mysticism, and especially Western modern mysticism, really is. Should mysticism be seen as based on a common transcendental ground that is shared by all religious traditions or is it bound by its co30ntext and thus specific to every tradition in which it appears (Hammersholt 2013; Kimmel 2008)? As seems to be clear, Eric Hermelin joins the discourse in which there are common transcendental grounds that connect different traditions of mysticism. But how? The answer seems to be found in another Swedish religious thinker that Hermelin has translated: Emmanuel Swedenborg.

When it comes to Swedenborg, there have already been discussions regarding Hermelin's comparison between Swedenborg and Persian Sufism in Swedish academia. Christian Dyresjö has performed an analysis of how these comparisons are made, using, among others, the concept of Influxus and the doctrine of correspondence (Dyresjö 2021). According to Dyresjö, Hermelin uses Swedenborg's doctrine of correspondence—which is built upon the thought of all physical things being in correspondence with the spiritual world—when commenting on Sufi poetry (Dyresjö 2021, p. 33). This could be some of the more direct connections Hermelin makes between the use of words in the Persian Sufi poems and the same or similar words in verses from the Old and New Testaments. What he does is not just a comparison of words—he puts a spiritual meaning in the use of these precise words. The words in different texts correspond to one another, and this corre000spondence is meaningful to Hermelin through his reading of Swedenborg.

Hermelin is also a proponent of more spontaneous commenting on the text, not always with comments that include references to other writers. Some of these comments are to contemporary debate in Swedish media—cultural or political. Other references are of a more local variety—referencing what someone said at public debates he has visited (Rumi 1934b, p. 135). There are also some comparisons in Hermelin's notes that are more general and cannot be accredited to the use of Swedenborg's doctrine of correspondence, described above. If Rumi writes about "the false people", Hermelin writes comments such as "the renaissance and the reformation" and when Rumi writes about "the blind", Hermelin comments "the atheists" (Rumi 1934b, pp. 67, 69). Hermelin must be aware that this is not what Rumi meant when writing these words, but he situates them in his own world as an explanation of what he claims has gone wrong with Christianity and in contemporary society. Such claims can be found in references that "the Swedish translation of Matthew 6:22 is a forgery"—without any further comments (Rumi 1933, p. 6). Or when Rumi mentions violence and oppression in some context, Hermelin makes references to "conscription" and "chemical warfare", thus relating Rumi's medieval text to the First World War and contemporary military techniques (Rumi 1933, p. 77). These references could be claimed to be more connected to Eric Hermelin himself, and his contemporary times, than to the actual original text.

### 3.3. Erik Hermelin in 20th Century Swedish Press

In relation to the end of the process described above, it is interesting to investigate the reactions to Hermelin's translation when they were published. What was written about Hermelin's works in the Swedish media during his lifetime? The writer and, from the 1970s forward, compiler of Hermelin's translations, Carl-Göran Ekerwald, claims that Hermelin introduced Sufism to the Swedish public (Hermelin 2007, p. 19). It is known that his works were held high by such writers as the eminent poet Vilhelm Ekelund (Ekelund 1945, p. 140). There are also a few reviews published. Ivan Harrie wrote in *Göteborgs Handels- och Sjöfartstidning* (Harrie 1931), regarding the release of the translation of Firdausi's *Shah-Namah*, that Hermelin is the translator behind a long row of books with unpronounceable titles published by Norstedt and Gleerups (two major Swedish publishing houses during this time). Unfortunately, without any success among the public, Harrie explains, these books are often found at book sales. According to Harrie, Islam has mostly created labyrinthine scholastic systems of low quality, these translations of Persian Sufi writings being an exception. Hermelin is claimed to have a crucial role in

presenting the Persian Sufis to the Swedish public and putting them into the right context together with Emmanuel Swedenborg and Jakob Böhme. Harrie also mentions Hermelin's use of contemporary political questions as a setting for certain verses. It seems like Harrie's admiration of Hermelin was answered. In the translations of *Mesnavi*, published two years after Harrie's article, Hermelin makes a reference to a text by Harrie in relation to Rumi (Rumi 1933, p. 181). Ellen Rydelius wrote a generally very enthusiastic article on Eric Hermelin's translations from Persian in *Svenska Dagbladet* (Rydelius 1937) and focused on the uniqueness of these translations. Hermelin is one of the few translators of medieval Persian poetry, and the only major one. Rydelius mentions the role of the woman in Islam and claims that the prejudice regarding Islam being a religion that is oppressive towards women is challenged by a Sufi such as Attar whose texts have been translated into Swedish by Hermelin. Part of the article is an interview with Hermelin who was close to his 80th birthday. Much focus is put on his early life, the period when he learned Persian while being in the British colonial army. Hermelin, who according to the article spent six years in India, is described as a globetrotter and expresses his positive views of the English and the British empire in the interview. When Eric Hermelin dies in 1944, a rather modest obituary is published in (Svenska Dagbladet. 11 October 1944). His relations and his unique role as a translator are described, but the travels are now described as research expeditions rather than as service in the British Imperial Army.

## 4. Encore—Encode!

Hermelin's influence continues after his death with the publication of parts of his work in *Persiska antologin—The Persian Anthology*. This compilation of Hermelin's works was released by the small publisher Bo Cavefors Förlag (BOC Förlag) in 1976, consisting of texts from Attar's *Tazkirat al-Awliyā* and texts by Rumi, Sadi, and Böhme, among others (Svensson et al. 2018, p. 143; Hermelin 1999). BOC Förlag was responsible for releasing a wide range of foreign authors translated into Swedish, among them Michel Foucault, Roland Barthes, and Ernst Jünger (Svensson et al. 2018, p. 14). Bo Caverfors Förlag was seen as the primary publisher of the radical left during its existence between 1959 and 1982, releasing texts by Mao Tse Tung, Joseph Stalin, and Kim Il Sung, among others. There was a scandal in the press when Cavefors released the collected texts of the still-active "Rote Armee Fraktion" (Svensson 2018). After the publisher went into bankruptcy, *The Persian Anthology* was rereleased by two other publishers, Näktergalen Förlag in 1986 and Ordfront Förlag in 1999 (Svensson et al. 2018, p. 143; Hermelin 1999). The editor of the anthology was Carl-Göran Ekerwald, a Swedish writer and critic. In Ekerwald's compilation of Hermelin's translation of Persian Sufi poetry, he added the translation of Jacob Böhme's *Vom übersinnlichen Leben* in its entirety because he believed it clarified Sufism (Ekerwald 2016, p. 15). But as we have noticed earlier this is a mixture that follows Hermelin's opinions and line of work well.

In 2007, another anthology, *Persiskt balsam* (*Persian Balm*) with translations made by Eric Hermelin and introductions written by Carl-Göran Ekerwald was released (Hermelin 2007). The content consists of texts by Sadi, Firdausi, Khayyam, and others. These are versions with very few comments by Hermelin—but the introductions by Ekerwald add some comparisons to Western thinkers. When describibng the Sufis, he compares them with the Greek Cynics and states that they actually "are Cynics" (Hermelin 2007, p. 14). Continuing his description of the Sufis, he discusses their principles of non-violence, comparing them to the Christian theological movement of Quietism, ending the comparison that Sufism "is Quietism" (Hermelin 2007, p. 19). Another comparison that he returns to several times is between Sufi poetry and the Swedish Academy member and poet Bertil Malmberg (Hermelin 2007, p. 288). It is possible to claim that Ekerwald continues in Hermelin's footsteps by putting Sufi poetry in a Western context, although he does this in a more subtle way than Hermelin did. The Hermelin-like comparisons continue in Ekerwald's own book about Rumi, *Vreden och begäret* (*The Rage and the Desire*) where he, following Hermelin, compares Rumi to Swedenborg using Swedenborg's idea of correspondence

between heaven and earth to explain Rumi's parable of the garden in *Mesnavi IV* (Ekerwald 2016, p. 38). This is the last book collecting Hermelin's translation so far. Interestingly enough, there seems to be an interest in Hermelin's work—and the person Eric Hermelin— from the contemporary Swedish identity/*Les Identitaires* movement. In 2014, a rather lengthy blogpost was published on the extreme right-leaning blog "Motpol" in which the, in the identity movement, rather established writer Joakim Anderson (more known under the pen-name "Oskorei") described Eric Hermelin's life and his translations. Also interesting is that both Hermelin's antiracism and his pro-Jewish stance during the 1930s are ignored in this article altogether (Andersen 2014).

## 5. Concluding Reflections

The main focus of this article, as formulated in the introduction, was to analyze Eric Hermelin's work and explore the process by which he created and delivered the texts of the Persian Sufi to a Swedish audience. Further, what could the consequences be when such delivery of a phenomenon to a cultural context is performed by a single individual? What happens with a phenomenon when its reception in a new cultural context is deeply connected to this individual and the compilations of his works? The encoding present in Eric Hermelin's translations of Persian Sufi puts the translated texts into the context of Sweden in the early 20th century, to Christianity and the translator's own life as a vagrant and a mental patient. This is done through constant references to text that are more related to the translator and his context than to the original translated texts. Most of these references could be classified, using my terminology in the analysis above, as "anachronistic", using examples from Hermelin's own lifetime to explain texts written several hundred years earlier. When it comes to references that are directed towards Christian mystics, such as Jacob Böhme and Emanuel Swedenborg, these references are less ad hoc, connecting both to the Swedenborgian concept of correspondence and to a debate within mysticism regarding how connected this strain of religion actually is to certain traditions, opening up for a more general mysticism less connected to different religious traditions. This encoding continues after Hermelin's death when compilations of his translations are published over the years. In Sweden during the 20th century, Hermelin's interpretation of the Persian Sufi is in a monopolistic position, with very small, we could almost claim no, opposition. This makes the encoding he added to his translations through different comments and references more or less without critique and secured him as the gatekeeper of Persian Sufism in Sweden.

**Funding:** This research received no external funding.

**Data Availability Statement:** Data sharing not applicable.

**Conflicts of Interest:** The author declare no conflicts of interest.

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
