# Peer review of "Baron Eric Hermelin—Translation and the Merge of Traditions; Encoding and Reception of Persian Sufi Poetry in 20th Century Sweden"

_religions, doi:10.3390/rel14040536_

Round 1

Reviewer 1 Report

Preferably, remove any personal reference in the article and 

line 25 'This is my starting point' = 'This serves as a starting point'

line 30 'S:t Lars' = St Lars [REMOVE :]

line 31 Ideally sentences do not start with a 'but'. I would rephrase something like this: 'Even from the hospital, he was able to release translations of Persian mystics such as Rúmi and Khayyam, along with translations of Christian mystics like Böhme and Swedenborg.'

line 34-35 remove personal reference. I would reword something like this: 'The main focus of this article will be on analyzing Eric Hermelin's work and exploring the process he used to create and deliver trans-traditional mysticism to a Swedish audience.'

line 37 I would rephrase and not start with an 'And'.

line 39 is superfluous.

lines 39-32 I would rephrase something like this: 'This article provides an overview of Eric Hermelin's work, but it should be noted that there is much more to be explored. The analysis presented here only covers a small portion of Hermelin's extensive corpus and is primarily based on examples. The purpose of this overview is to lay the groundwork for further research in this area.'

line 44 This paragraph could be split in two or three parts. It is too long as it is.

line 45 i would rephrase 'by a family'. lines 44-46 could be rephrased in the following manner: 'Following a brief stint as a student at Uppsala University, he was sent abroad by his family who were concerned that his unconventional behavior and penchant for heavy drinking would lead to scandal. (Lindahl 46 1982, p. 39).'

line 51 'S:t Lars' = St Lars [Remove :]

line 54 'he had a possibilities' = 'he had possibilities' [Remove a]

line 56 ; = .

line 58 were = was

line 59 'both archbishop' = 'both the archbishop' [insert 'the']

line 64 1940:s = 1940s or 1940's

line 70 west = West

line 72 Remove 'Eric'

line 73 Do not start with an 'And.'

line 76-77 rephrase and can could become this: 'During the First World War, he was supportive of Germany. However, his views evolved over time.'

line 77 1930:s = 1930s or 1930's

line 78 1930:s = 1930s or 1930's

line 78 'especially the antisemitism' = 'especially with the rise of antisemitism'

line 85 ' more or less' is superfluous. 

line 84 remove personalisation. I would rephrase something like this: lines 84-87: 'Taking into account Eric Hermelin's unique worldviews and opinions, as well as his prominent position as the primary translator of the Persian Sufi corpus in Sweden, he can be seen as a gatekeeper for the Swedish reception of Persian Sufism.'

line 89 I think there could be a better subheading. But i'll leave it up to the author. Perhaps: 'Information Flow and gatekeeping'

line 91 remove 1st person pronoun. I would formaulate something along these lines: 'A primary theoretical foundation can be called "Information flow," and it is based on a blend of information flow analysis and process management commonly utilized in the field of Archival Science.'

line 96 remove personal reference ('us'). I would rephrase something like this: 'Information flow aids in answering questions regarding how information entities move from one location to another.'

line 98 remove personalisation (we) in an academic article.

line 99 remove personalisation (our)

107 I would start a new paragraph. Avoid long paragraphs.

111 this sentence needs to rephased: Nobody else did this – or at least they did not get published. 

line 126 'process that are' = 'process that is'

line 127 remove personalisation

line 128 remove personalisation. something like this: 'The process being described here encompasses more than just a single organization and involves both smaller agents (individuals) and larger entities (publishers, magazines).'

line 135 ; = : or ,

line 135  The Environment = the Environment

line 137-140 rephrase to remove personalition: 'The environment in which Eric Hermelin carried out his translations differed significantly from the context of the present day. It could be argued that these early threads of our current information society are present in Hermelin's work. The emergence of a form of globalism is evident, wherein an individual, by aligning themselves with the colonial powers as Hermelin did, was able to travel the world.'

line 146 movement of information were = movement of information was

line 155-156 I would rephrase as such: 'There are additional factors at play that position Eric Hermelin as a gatekeeper of Persian Sufi poetry within the Swedish market.'

line 158 remove personalisation [we]

line 159-161 I would rephrase like this: 'In modern times, the same devices utilized for daily tasks such as paying bills, watching television, and working can also provide access to texts from around the world.'

lines 163-164 I would rephrase something like this to remove personalisation: 'This characteristic is what positions him as a gatekeeper, and the following section will provide an explanation of what gatekeeping entails in this particular context.'

line 165 remove personalisation

lines 165-170 need to be rephrased.

line 170 has already been asserted in the article. No need to keep on repeating this.

lines 174-177 i would rephrase like this: 'Based on the sources available during the research, it appears that Eric Hermelin was the sole translator and commentator of Persian Sufi texts, as no other major translator has been identified through searches of the National Library Catalog and contemporary news media.'

line 178 is this sentence worthy of an academic work? 

line 179 remove personalisation [we]

line 184 Start a new paragraph with 'Stuart Hall'

line 201 remove personalisation [I]

line 204 'I will also show' = 'the article seeks to show'

line 210 remove personalisation [my]

line 218 hight = high

line 224 please do not start sentences with And or Buts. English discourages it.

consider splitting up this section. It has a very long paragraph.

line 239 remove personalisation. rephrase.

line 248 remove personalisation. rephrase.

line 253 insert space 'text,rather'

line 272 ad hoc here is in italics. But not the same in line 255.

line 272 remove personalisation. rephrase.

line 274 remove personalisation [we]

line 291 'All in all' is superfluous. It has been used elswhere too. I would omit it.

line 364 'is. Is' Avoid this 

line 473 .. = .

line 477 1930:s = 1930s or 1930's

line 488 ad hoc in italics or not?

line 480-481 'puts the translated texts into a context' ... what does this mean?

Author Response

Thank you for your very helpful remarks - this will help me a lot.

Reviewer 2 Report

The author’s prose style leaves much to be desired. A few examples may suffice:

where he eventually joined the Imperial army and were shipped to India, and finally to Australia, where he apparently behaved in such ways that he were sent home (l. 48-50)

he had a possibilities to pick up the language (l. 54)

no other language in the west have so much Persian Sufi (l. 70)

But his views changed over time; In the 1930:s (l. 76-77)

This could some of the more direct connections Hermelin does between the use of words in the Persian Sufi poems and the same, or similar words in verses from the old and new testament. (l. 377-9)

Section 1. Introduction (l. 21-88) provides biographical information on Hermelin, as well as the author’s central line of investigation: “In this article I will analyze Eric Hermelin’s work and focus on the process of creating and delivering a trans-traditional mysticism to a Swedish audience.” (l. 34-5) This line of investigation gives rise to two specific research questions: “What could the consequences be when such delivery of a phenomenon to a cultural context is performed by a single individual? And what happens with a phenomenon when it’s reception in a new cultural context is deeply connected to this individual and the compilations of his works?” (l. 35-38) The introduction succeeds in presenting the author’s argument and choice of focus.

Section 2. Information Flow and the role of the gatekeeper (l. 89-213) outlines the author’s theoretical and methodological approach. The author’s point of departure is the concept of “information flow” and reception theory. The author focuses on the concept of gatekeeping, describing Hermelin as the only translator of Persian poetry into Swedish active at this time. The concept of information flow is tied to Hermelin’s biography, and the author argues that Hermelin’s travels as a participant in the British Empire allowed Hermelin access to flows of information that would otherwise have been unobtainable. The author makes use of reception theory by “analyzing reviews and articles” (l. 182) and by “analyzing the encoding in which Sufi poetry was served to the Swedish public” (l. 183). The concept of encoding is used, following Stuart Hall, in the sense of “the context and the ideology added to the content by the publisher” (l. 187).

Section 3. Consequences of Eric Hermelin as a gatekeeper (l. 214-433) is divided into an examination of Hermelin’s translation of Firdausi’s Shah-Namah and Jalal al-din Rumi’s Mesnavi, which is followed by a brief discussion of “Erik Hermelin in 20th century Swedish press”. The purpose of the section is unclear. If the author’s intent is to highlight the “consequences” of Hermelin’s role as gatekeeper, these consequences are not made clear. The examination of Hermelin’s two translations is a study of how Hermelin chooses to annotate his translations. This line of commentary is relevant to an understanding of Hermelin as a translator, but they are more properly in line with the discipline of translation studies, a field of study which the author does not reference. If the purpose of the discussion of Hermelin’s translations is to shed light on his role as gatekeeper, as stated, then this purpose is not achieved.

The singe paragraph on “Erik Hermelin in 20th century Swedish press” is a short overview of the critical reception of Hermelin’s translations. This overview does not contribute any information that would improve our understanding of Hermelin’s work as a translator. This section appears to be the author’s attempt at employing reception theory, as noted in section 2, but if so, the author should either commit to using reception theory throughout the entirety of the article or choose another methodology.

The description of Frithiofs saga as an example of “epic Swedish poetry” (l. 308) is puzzling. Is the author referencing the old Norse saga, or Esaias Tegnér’s modern retelling?

The discussion of Fröding (l. 321-7) does not discuss Fröding’s interest in esotericism/mysticism, and also does not account for the circumstances driving Fröding’s lapse into illness, specifically the charges of indecency brought against him in court.

The discussion of Vernon Lee is short and speculative (l. 331-8), and the author’s suggestion that “Vernon Lee was a critic of modernism and modern life, which might be a reason for [Hermelin] both to read her work and use parts of her text in his comments” should either be removed or developed.

Section 4. Encore – Encode! (l. 434-478) provides a cursory overview of the afterlife of Hermelin’s translations. The section does not connect to the author’s stated intent, and offers almost no information that would enable us to better understand either Hermelin’s work as a translator or his role as gatekeeper and participant in flows of information.

Section 5. Concluding Reflections (l. 479-96) does not accurately reflect the stated intent of the article. The author at first expresses an intent to examine “the process of creating and delivering a trans-traditional mysticism to a Swedish audience” (l. 34-5). This goal is not achieved. The central concept of a “trans-traditional mysticism” is not explained, and the impact of this mysticism in Sweden is not explored. The author’s research questions (l. 35-8) remain similarly unanswered.

The list of references does not follow any recognizable format. Book titles are not given in italics. Page numbers are not provided for journal articles.

Author Response

Thank you - You gave me several important remarks to think about.

Reviewer 3 Report

This is a fascinating article, which I very much enjoyed reading. The author presents a clear and compelling argument and shows great insight into the life and work of this interesting translator and mediator of Persian poetry. This piece should absolutely be published. 

The article does, however, need some English editing throughout. While the author's English is otherwise excellent and expressive, there are quite a few small grammatical errors and incomplete sentences throughout. These can easily be fixed, however. One tiny additional correction: The author Vernon Lee was not American, but rather British. 

Author Response

Thank you so much for your kind remarks. I will change the nationality of Vernon Lee, I don't know how I got that wrong.

Round 2

Reviewer 1 Report

line 11 - thoughts ARE analyzed (not IS analyzed)

Author Response

Thank you - I will change that.

Reviewer 2 Report

The author has improved the article to such an extent that it can be published.

Author Response

Thank you!